# A Novel Fluorescent Sensor Based on Aptamer and qPCR for Determination of Glyphosate in Tap Water

**DOI:** 10.3390/s23020649

**Published:** 2023-01-06

**Authors:** Yong Shao, Run Tian, Jiaqi Duan, Miao Wang, Jing Cao, Zhen Cao, Guangyue Li, Fen Jin, A. M. Abd El-Aty, Yongxin She

**Affiliations:** 1Institute of Quality Standardization & Testing Technology for Agro-Products, Chinese Academy of Agricultural Sciences, Beijing 100193, China; 2Key Laboratory of Agrofood Safety and Quality (Beijing), Ministry of Agriculture and Rural Areas, Beijing 100081, China; 3State Key Laboratory for Biology of Plant Diseases and Insect Pests, Institute of Plant Protection, Chinese Academy of Agricultural Sciences, Beijing 100193, China; 4State Key Laboratory of Biobased Material and Green Papermaking, Shandong Academy of Sciences, Qilu University of Technology, Jinan 250353, China; 5Department of Pharmacology, Faculty of Veterinary Medicine, Cairo University, Giza 12211, Egypt; 6Department of Medical Pharmacology, Medical Faculty, Ataturk University, 25240 Erzurum, Turkey

**Keywords:** glyphosate, aptamer, qPCR, SYBR Green I, sensor

## Abstract

Glyphosate (GLYP) is a broad-spectrum, nonselective, organic phosphine postemergence herbicide registered for many food and nonfood fields. Herein, we developed a biosensor (Mbs@dsDNA) based on carboxylated modified magnetic beads incubated with NH_2_-polyA and then hybridized with polyT-glyphosate aptamer and complementary DNA. Afterwards, a quantitative detection method based on qPCR was established. When the glyphosate aptamer on Mbs@dsDNA specifically recognizes glyphosate, complementary DNA is released and then enters the qPCR signal amplification process. The linear range of the method was 0.6 μmol/L–30 mmol/L and the detection limit was set at 0.6 μmol/L. The recoveries in tap water ranged from 103.4 to 104.9% and the relative standard deviations (RSDs) were <1%. The aptamer proposed in this study has good potential for recognizing glyphosate. The detection method combined with qPCR might have good application prospects in detecting and supervising other pesticide residues.

## 1. Introduction

Glyphosate (GLYP), with the chemical name *N*-(phosphoryl-methyl) glycine, is a nonselective, broad-spectrum organic phosphine herbicide used in agriculture to eliminate various annual and perennial weeds [1,2,3,4] around the globe. Following application, it is evenly distributed and rapidly translocated to regions of active growth within the plant, causing death. However, the high herbicidal activity led to the abuse of glyphosate [5,6]. Many studies have proven that glyphosate has toxic effects and potentially harms the ecological environment and human health [7,8,9]. Furthermore, residues have been found in soil, even groundwater, due to extensive usage. Moreover, recent studies have shown that glyphosate accumulation in the environment may cause a certain degree of reproductive toxicity, teratogenicity, mutagenicity and carcinogenicity in humans [10,11]. To reduce the impact of glyphosate residues on food safety and human health, detecting glyphosate residual levels is crucial and essential.

As concerns and studies on the behavior of glyphosate in plants and the environment are growing, several methodologies, including high-performance liquid chromatography (HPLC) [12,13], gas chromatography-mass spectrometry (GC–MS) [14,15,16], LC–tandem mass spectrometry (LC-MS/MS) [17,18], liquid chromatography (LC) [19,20], ion chromatography (IC) [21,22], capillary electrophoresis (CE) [23,24], enzyme-linked immunoassays (ELISA) [25,26], fluorescence detection [27,28] and electrochemical luminescence methods [29,30], have been developed to detect its residues. Although the sensitivity and specificity of these methods are relatively high, their shortcomings are also apparent. They usually require expensive instruments, professional operators, time-consuming sample pretreatment and high testing costs, with certain limitations for glyphosate detection. Therefore, developing a rapid, simple, inexpensive, high-sensitivity and high-specificity sensor is crucial for quantitatively detecting pollutants.

Aptamers are nucleic acid molecules synthesized in vitro by a process known as Systematic Evolution of Ligands Exponential Enrichment (SELEX), which has unique binding properties to different targets, such as proteins, small molecule metal ions and whole cells [31,32,33,34]. As a new type of molecular probe, it can be synthesized with high purity in vitro. The cost is lower, which is a point of superiority compared with antibodies. After the aptamer sequence is determined, it can be synthesized with high reproducibility and high purity from commercial sources at a lower cost, which is an advantage compared with antibodies. Antibodies generally take 3–6 months to prepare and synthesize, while aptamers only take a few hours to several days to synthesize. The aptamer has high thermal stability, low immunogenicity, and chemical stability. In addition, the preparation process of antibodies is usually very long, while the synthesis time of aptamers is usually shorter. At the same time, it can meet the experimental design requirements of modification by chemical labels, such as nanoparticles, fluorescent groups and functional groups. It will not affect the affinity between them and the target molecules [35,36,37]. They are generally more stable under harsh conditions than antibodies [38,39,40]. After thermal denaturation, the aptamers usually return to their native state. Furthermore, they can be used as effective recognition elements in biosensors and are widely used in the detection field for biosensing. Consequently, some results have been achieved in aptamer-based pesticide detection technology. For instance, what has gained more application in sensing is generally color-based detection, which has more directly detectable results [41]. Aptamer sensors have been reported to detect fipronil, non-nitrothion and diazinon in fruit and vegetable samples. These aptamer sensors have the advantage of being highly sensitive and specific [42,43,44].

SYBR Green real-time qPCR can monitor the amplification process in real time by detecting the fluorescent signal emitted by a dsDNA-specific dye (SYBR Green I). In real-time fluorescent quantitative PCR amplification, the fluorescent group can monitor the amplification products in each cycle of PCR in real time and the operation is convenient and straightforward. DNA amplification and data analysis can be carried out in a closed system under the same conditions so that samples and products cannot be polluted. Electrophoresis confirmation of standard PCR products can be directly omitted. To the best of our knowledge, there are few studies on the detection of glyphosate based on the combination of aptamers and qPCR. Chen et al. developed a time-resolved luminescence assay for glyphosate based on G-quadruplexes but not in combination with qPCR. Therefore, the aptamer sequence from this article was used. Herein, aptamer-SYBR Green I-based sensing technology was established to rapidly detect glyphosate by qPCR using the single-stranded DNA released from the aptamer after recognizing glyphosate. The recognition process of the aptamer for glyphosate was converted to the qPCR cycle threshold (CT) value.

## 2. Experiment

### 2.1. Materials and Instruments

All DNA sequences were synthesized and purified by Sangon Biotech (Shanghai) Co., Ltd. (Shanghai, China) The sequences of the DNA strands is shown in the Appendix A. Carboxyl magnetic beads were procured from Sangon Biotech (Shanghai) Co., Ltd. Power Up™ SYBR™ green premix was acquired from Thermo Fisher Scientific (Waltham, MA, USA). Glyphosate, acetamiprid, chlorpyrifos, dimethoate, trichlorfon, methomyl and propoxur (purity greater than 98%) standards were provided by Dr Ehrenstorfer GmbH (Augsburg, Germany). Other chemical reagents were purchased from Sinopharm Group Co., Ltd. (Beijing, China). All purchased chemicals were used as received without further purification. The water involved in the experiments was purified by Milli-Q (Merck & Co., Rahway, NJ, USA) with a resistivity of 18.2 MΩ cm).

UV-2600 (SHIMADZU (CHINA) Co., Ltd. (Beijing, China)); TECAN Infinite 200 PRO Multifunctional Enzyme Reader (Männedorf, Switzerland); 7500 real-time fluorescence quantitative PCR instrument, Thermo Fisher Scientific (Waltham, MA, USA); and HM100-Pro, Dragon Laboratory Instruments Limited (Beijing, China) were used in this study.

Acetamiprid, chlorpyrifos, dimethoate, trichlorfon, methomyl and propoxur were separately prepared (4.5 mol/L, 0.29 mol/L, 4.4 mol/L, 3.9 mol/L, 0.62 mol/L and 4.8 mol/L) with methanol, diluted to 0.045 mol/L, 0.00029 mol/L, 0.044 mol/L, 0.039 mol/L, 0.00062 mol/L and 0.048 mol/L with deionized water. Glyphosate was prepared at 0.006 mol/L with deionized water and finally diluted to 0.03 mol/L, 0.012, 0.006, 0.003, 0.0012 and 0.0006 mol/L with a deionized water gradient.

### 2.2. Magnetic Bead Activation

The procedure used to activate magnetic beads was performed according to the manufacturer’s protocols (Sangon Biotech Co., Ltd., Shanghai, China). After the carboxyl magnetic beads were whirled for 1 min, 100 μL was placed into a 2 mL centrifuge tube. Next, the supernatant was removed after magnetic separation and then 100 μL 25 mM MES solution was added and whirled for 15 s. After magnetic separation, the supernatant was removed and washed three times with deionized water. After that, 500 μL (10 mg/mL) 1-ethyl-3-(3′-dimethylaminopropyl) carbodiimide (EDC) and 100 μL (10 mg/mL) *N*-hydroxy-succinimide (NHS) were quickly introduced into a centrifuge tube. After mixing, the mixture was shaken slowly at room temperature for 30 min and then stored at 4 °C.

### 2.3. Preparation of Mbs@dsDNA and Pre-Experiment

Then, 600 μL of activated magnetic beads was taken, the supernatant was discarded after magnetic response, 600 μL of 10 μM polyA solution was added, thoroughly mixed and incubated in a 37 °C water bath overnight. The following day, the supernatant was removed after magnetic separation, redissolved in 600 μL deionized water and stored at 4 °C.

To obtain purified Mbs@polyA after magnetic separation, that polyA not combined with the magnetic bead is removed from the supernatant. 600 μL of 2 μmol polyT-aptamer solution was denatured at 90 °C for 2 min, then slowly dropped to room temperature and added to Mbs@polyA After vortex oscillation in solution for 2 min, it is slowly shaken at room temperature for at least 30 min. Unhybridized polyT aptamer in the supernatant was removed after magnetic separation. Then, 600 μL of 2 μmol complementary DNA (C-DNA) solution was denatured at 90 °C for 2 min, slowly dropped to room temperature and added to the above solution. After 2 min of vortexing, the mixture was shaken slowly at room temperature for at least 30 min. Then, magnetic separation was performed to remove the nonhybridized C-DNA from the supernatant. After washing with deionized water three times, Mbs@dsDNA was suspended in deionized water at 4 °C for standby.

Twenty microliters of 0.6 μmol/L–60 mmol/L glyphosate and blank samples were added to the prepared Mbs@dsDNA system and incubated at room temperature for 3 h. After magnetic separation, the supernatant was taken for qPCR and photographed under a UV lamp.

### 2.4. Specificity Detection

A series of centrifugation tubes containing 60 μL of Mbs@dsDNA solution was prepared. After magnetic separation, the supernatant was removed and 60 μL of different solutions, including 0.045 mol/L acetamiprid, 0.00029 mol/L chlorpyrifos, 0.044 mol/L dimethoate, 0.039 mol/L trichlorfon, 0.00062 mol/L methomyl, 0.048 mol/L propoxur and 0.006 mol/L glyphosate, were added to each tube and incubated at room temperature for 3 h. After incubation, the supernatant was separated by magnetic separation and the released C-DNA in the system was identified by UV spectrophotometry and then detected by qPCR. qPCR was carried out in a total volume of 20 μL in the quantitative study. The conditions of amplification reactions are shown in the Appendix A.

### 2.5. Determination of Glyphosate

A series of centrifugation tubes with 60 μL of Mbs@dsDNA in each tube were prepared. After magnetic separation, the supernatant was removed and 60 μL glyphosate at different concentrations of 0.03, 0.012, 0.006, 0.003, 0.0012 and 0.0006 mol/L was added into each tube.

After 3 h of incubation, the supernatant was separated by magnetic separation and detected by UV spectrophotometry and qPCR.

### 2.6. Actual Sample Testing

Tap water from the city of Beijing was taken as the actual sample and glyphosate at concentrations of 8 μmol/L, 4 μmol/L and 2 μmol/L was added according to the national standard of China (GB/T 5749-2006). Each sample was tested 3 times. The relative standard deviation (RSD) of the test results was used to evaluate the method’s precision.

## 3. Results and Discussion

### 3.1. Experimental Principle

Figure 1 shows a schematic diagram of the glyphosate detection method based on qPCR signal amplification proposed in this paper. As shown in Figure 1A, polyA is formed after incubation and connection with magnetic beads Mbs@polyA. Subsequently, the aptamer was hybridized with the polyT-modified aptamer (polyT aptamer) according to the principle of complementary base pairing in DNA molecules. The aptamer was assembled on the magnetic bead.

Afterwards, the DNA complementary to the aptamer (C-DNA) is assembled on the magnetic bead through hybridization with Mbs@dsDNA. Through magnetic separation, redundant poly T-aptamers and C-DNA without double strands can be eliminated. As Figure 1B shows, when glyphosate is present in the system, the aptamer can specifically recognize glyphosate and form Mbs@Aptamer@GLYP. The binding force between the aptamer and glyphosate is stronger than that between the aptamer and C-DNA. With the help of an external magnetic field, the C-DNA and fMbs@Aptamer@GLYP were separated. The supernatant containing C-DNA was extracted and utilized in qPCR as a template. SYBR Green I was used to record the fluorescence signal obtained by qPCR and the linear equation was established by the logarithm of the CT value and concentration. Then, the content of glyphosate was calculated and the amount of glyphosate was calculated. When other interfering pesticides and blanks are added to the system, C-DNA will not be separated from the Mbs@dsDNA and the signal will not appear in qPCR.

### 3.2. Characterization of Mbs@dsDNA and Aptamer Sensor Feasibility Validation

The prepared Mbs@dsDNA in each step was characterized by ultraviolet and visible light and glyphosate was detected and characterized by fluorescence photography. When the peak value had a significant difference at 260 nm, the Mbs@dsDNA preparation was successful. When glyphosate is detected, the dye SYBR Green I can only be embedded in the double helix structure of DNA, so the fluorescence signal will be sent out when glyphosate is present in the system. As shown in Figure 2D, SYBR I can be embedded in the double helix structure of DNA and emit fluorescence. When there is no glyphosate in the system, SYBR Green I cannot embed into the double helix structure of DNA and will not emit a fluorescence signal because qPCR cannot be successfully carried out to obtain double-stranded DNA. In contrast, when glyphosate is present in the system, the aptamer specifically binds to it and releases C-DNA into the qPCR to successfully amplify double-stranded DNA. SYBR I is embedded in the DNA double helix structure and emits a fluorescence signal under the excitation of ultraviolet light.

### 3.3. Validation of the Specificity of the Aptamer Sensor

The specificity of the test method was evaluated and the selected pesticides and glyphosate were tested under the same experimental conditions. The specificity was identified by observing the absorbance change of the supernatant and whether the qPCR could obtain the fluorescence signal. As shown in Figure 3A, the addition of other insecticides cannot trigger the binding of the aptamer to them, so the absorbance of the supernatant has not changed, indicating that the detection method has good specificity for glyphosate. As shown in Figure 2B, a strong fluorescence signal can only be generated when glyphosate is detected. As mentioned above, the specificity of the test method is reasonable.

### 3.4. Glyphosate Detection and qPCRs Based on Mbs@dsDNA

In the presence of glyphosate, the C-DNA produced by its specific recognition by Mbs@dsDNA enters the qPCR to obtain a CT value. The CT value is related to the concentration of glyphosate. Even though the difference in absorbance in the supernatant was insignificant, the higher the concentration of glyphosate, the higher the CT value produced. Therefore, CT values are positively correlated with the concentration of glyphosate. Figure 4A,B shows that the absorbance in the supernatant and the CT value increased with increasing glyphosate concentration. There was a good linear relationship between the CT values and the logarithm of the concentration in the range of 0.1–5 ppm glyphosate. The linear equation is y = −4.0799x + 26.20509. The linear correlation coefficient is *R*^2^ = 0.99742 and the detection limit obtained by the experiment can reach 0.6 μmol/L; the national standard value is 3 μmol/L, which is far lower than the maximum residue limit of glyphosate in the national standard.

### 3.5. Application of This and Other Methods to the Detection of Glyphosate in Real Samples

To further verify the practicability of the method in the actual scene, tap water spiked with glyphosate was used as a real sample for detection. The validity of the method was confirmed by the reasonable range of recovery (03.4~104.9%) and relative standard deviation (0.4~0.73%) in the Appendix A. To illustrate the method’s ability to detect glyphosate, a crosswise comparison was carried out among all sorts of available technologies in Table 1. Different detection methods detect different kinds of actual samples, so the LOD is different. As shown in Table 1, glyphosate is primarily detected in agricultural products and tap water. Tap water is used as the real sample model in this experiment. The results are consistent with other detection methods, indicating that the developed adaptive sensor has good accuracy and is superior to some detection methods.

## 4. Conclusions

In conclusion, by utilizing the specific recognition function of the aptamer, a biosensor for the recognition of glyphosate based on the aptamer, magnetic beads and qPCR was invented. The linear range of the method was 30 mmol/L to 0.6 μmol/L and the lowest detection limit detected by the experiment was 0.6 μmol/L. The detection technology can be applied to detect glyphosate in actual samples (such as tap water) and has good potential for application. However, the method can still be further reduced in terms of detection limit and errors due to sampling pretreatment. Therefore, when these problems are solved in the future, it will be more beneficial for other pesticides.

## Figures and Tables

**Figure 1 sensors-23-00649-f001:**
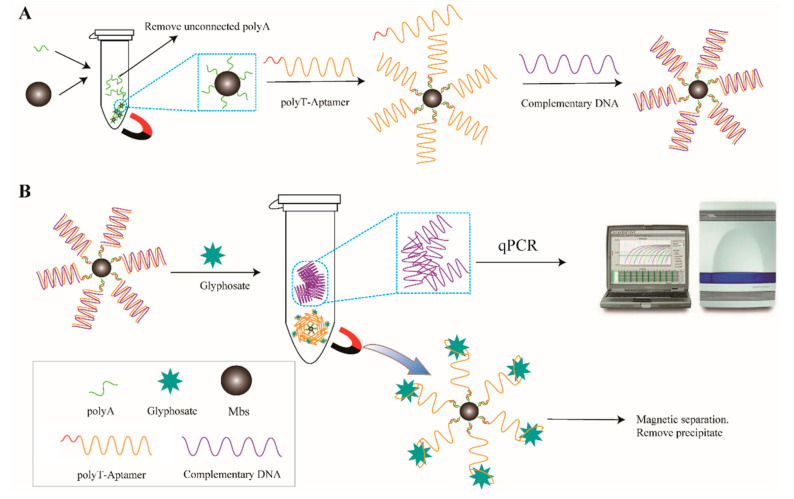
(**A**) Forming dsDNA compound. (**B**) Detection of glyphosate by qPCR. Aptamer sequence: GCTAGACGATATTCGTCCATCCGAGCCCGTGGCGGGCTTTAG GACTCTGCGGGCTTCGCGGCGCTGTCAGACTGAATATGTC.

**Figure 2 sensors-23-00649-f002:**
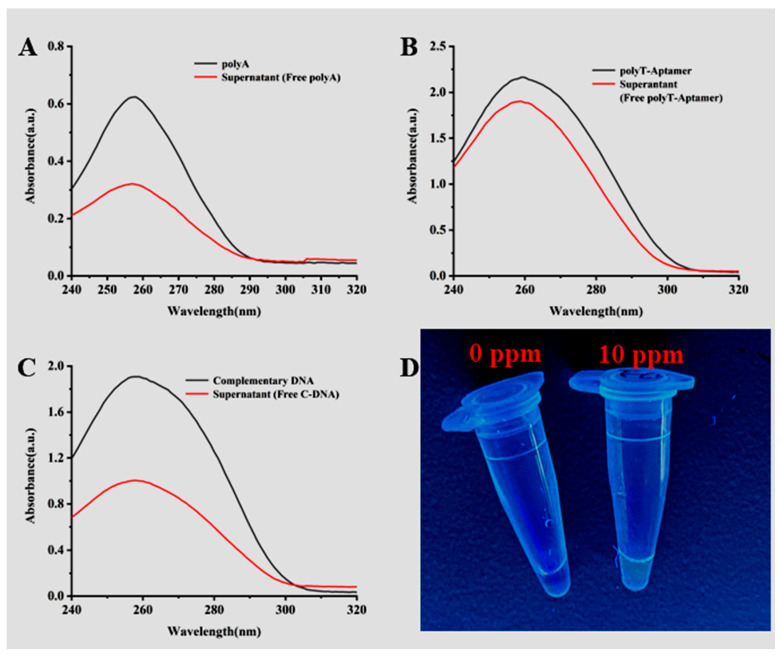
(**A**) UV-Vis absorption spectrum: polyA (black), supernatant (free polyA) (red), (**B**) UV-Vis absorption spectrum: polyT-Aptamer (black), supernatant (free polyT-Aptamer) (red), (**C**) UV–Vis absorption spectrum: complementary DNA (black), supernatant (free C-DNA) (red), (**D**) Fluorescent photos under a 365 nm UV lamp.

**Figure 3 sensors-23-00649-f003:**
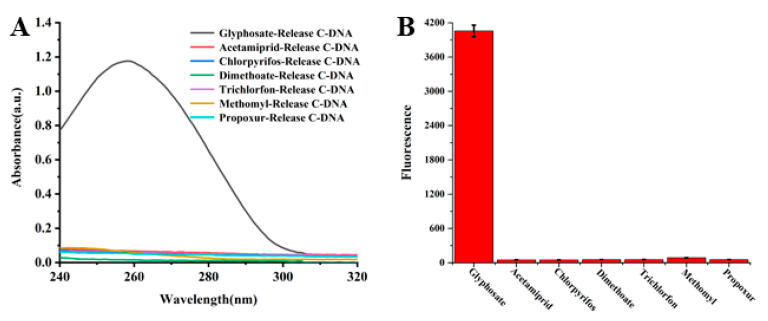
(**A**) Different pesticides and Mbs@dsDNA UV–Vis absorption spectrum of C-DNA released after incubation; (**B**) Fluorescence intensity histogram at 520 nm.

**Figure 4 sensors-23-00649-f004:**
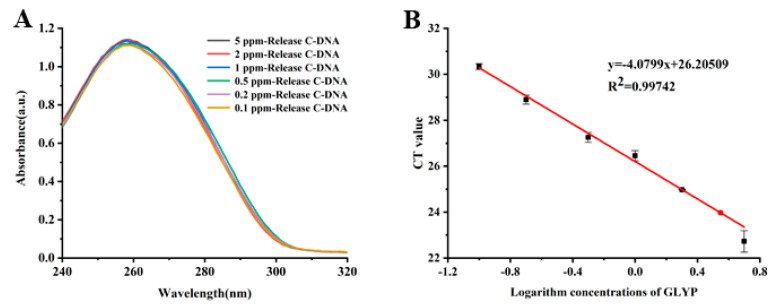
(**A**) Different concentrations of glyphosate and Mbs@dsDNA UV-Vis absorption spectra of C-DNA released after incubation. (**B**) Standard curve of the CT value and logarithm concentration of glyphosate.

**Table 1 sensors-23-00649-t001:** Performance evaluation of glyphosate analysis technology.

Method	LOD	Recovery	Sample	Reference
GC–MS	3 μmol/L	96.7–107.7%	Serum and urine	[45]
ELISA	0.047μmol/L	87.4–103.7%	River water, Tea and soil	[46]
HPLC	0.004 μmol/L	80.1–109.4%	Natural water	[47]
LC	6 μmol/L	80.63–97.11%	Soil	[48]
IC	30 μmol/L	80–110%	Honey	[49]
CE	800 mmol/L	89.4–93.7%	Hemp	[50]
LC–MS/MS	6 μmol/L	97–110%	Breast milk	[51]
EC	0.16 mmol/L	72.7–98.96%	Cucumber Tap water	[52]
qPCR	0.6 μmol/L	103.4–104.9%	Tap water	This work

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
