# Peer review of "A Novel Fluorescent Sensor Based on Aptamer and qPCR for Determination of Glyphosate in Tap Water"

_sensors, 2023, doi:10.3390/s23020649_

Round 1

Reviewer 1 Report

This paper described a novel fluorescent sensor based on aptamer and qPCR for determination of glyphosate in tap water. There are many typographical errors in this paper. The authors should modify. There are many papers on aptasensor for glyphosate (DOI: 10.1016/j.snb.2021.130288、10.3390/bios12110920、10.1021/acsagscitech.1c00147). You should clearly describe novelty of the paper. Line 126: The information on the aptamer is necessary. Did you screen the aptamer by yourself? Line 135: Change "20" to "twenty". Lines 161 to 178: The part of Experimental principle should be move between 2.1. Materials and instruments and 2.2. Magnetic bead activation. Line 174: PCR of C-DNA is done in a crude sample solution. If so, you don't need to use Mbs@dsDNA. dsDNA is enough to detect glyphosate. Table 1: Unit of LOD should be unified.

Reviewer 2 Report

Although the research article is a significant assay for glyphosate, it needs careful editing for scientific concepts and English grammar.

The concentration for all chemicals must be provided in molarity for consistency with the contemporary style.

The explanation provided for Figures 1, 2, 3, and 4 needs to be elaborated. Please provide detailed captions for each figure with a brief and concise explanation of the content of the figure.

The introduction is too short. Moreover, a separate discussion is well-deserved for this article. However, if authors wish to combine results and discussion, they need to provide in depth discussion of previous literature in the currently used style.

Authors should compare aptasensors for glyphosate (herbicide) with other aptamers such as commonly used pesticides in agriculture. I recommend a recent review article: https://doi.org/10.1016/j.teac.2022.e00184 and few studies by the group on malathion, fipronil, diazinon, and fenitrothion, that could be compared with aptamer-based herbicide sensing.

Please provide details of the methodology or provide valid references from where methods are adopted.

The subheadings of the result section need revision, and each title should tell more information of the results being discussed.

Reviewer 3 Report

The study developed a new glyphosate detection method by leveraging aptamer, magnetic beads, and qPCR, with high specificity and sensitivity. The output of the study should be interesting to readers in related fields. Overall, more detailed information is needed for the introduction of prior work and description of the experiments / results (including figure legend). In addition, the language / wording of the manuscript needs to be refined. More specific suggestions for revision are listed below.  

Point-to-point Comments:

·       Line 55: “They normally required expensive instruments, professional operators, time-consuming sample pretreatment and high testing cost…” – suggest providing more nuanced details. E.g., what are the specific limitations for each method. Also, try to quantify them (e.g., time-consuming – how much time?)

·       Line 64: “the cost is lower, which is a point of superiority compared with antibody” – should quantify. Same comment for the preparation time comparison

·       Line 83: “there are few studies on the detection of glyphosate based on the combination of aptamers and qPCR.”—So, are there any prior examples about aptamers being used for detection of glyphosate (either with or without combination with qPCR)? If so, need to overview them with sufficient details in the Introduction section.

·       Figure 1B - the key for “polyT-Aptamer” in the box is wrong

·       In preparation of Mbs@dsDNA, Is it necessary to anneal poly-A to Poly-T-aptamer first, and then hybrid the C-DNA? Is that feasible to anneal the 3 all together in a single step? The authors should comment on this in the text

·       Need more description about how you found/designed the sequence of the aptamer? (maybe add to the Introduction section). Also, the authors should report the sequence of the Aptamer in the main text (maybe the first sub-section of “results and discussion”)

·       Line 174: “The fluorescent signal obtained from qPCR with SYBR Green I was recorded to set up the linear equation.” – need to make it clear that the “linear correlation” refers to either “fluorescent signal intensity (at a given number of PCR cycle) vs. glyphosate concentration” or “CT value from qPCR vs. log of glyphosate concentration”.

·       Line 187: “indicating that polyA is successfully bound to the magnetic beads to obtain the composite Mbs@polyA” – polyA loss during the wash step is not taken considered. Same comment for the second hybridization step.

·       Line 200: “the qPCR reaction cannot be carried out when there is no glyphosate in the system, so SYBR Green I cannot embed the double helix structure of DNA and does not emit fluorescence under UV light.” – I understand what you meant to say. But the language needs to refine to make it more accurate. Also, for Fig 2D, need to specify number of PCR cycles you conducted for the two samples.

·       Line 221: “According to the pre-experiment, the feasibility and specificity of the method is verified.” – The specificity has not yet been verified (by line 221).

·       Figure 3A – was the UV absorption measured before PCR or after? Need to make it clear.

·       Need to discuss the perceived limitations of the probing method presented in this work.

·       Line 248 “which is far lower than the maximum residue limit of glyphosate in the national standard.” – should just say what the national standard value is in the text.

·       Line 254: “The validity of the method was confirmed by the reasonable range of recovery [103.4% ~ 104.9%]” – this range here is contradictory with that shown in Table 1.

·       Line 257: “a crosswise comparison was carried out among all sorts of available technologies in Table 1.” – the authors should also summarize key insights from the table (comparison with prior work) in addition to showing the data.

Round 2

Reviewer 1 Report

I have confirmed that all points that all reviewers pointed out to be revised were thoroughly revised by authors.

Reviewer 2 Report

I believe the manuscript has been sufficiently improved to warrant publication in Sensors.